# Distinct Postprandial Bile Acids Responses to a High-Calorie Diet in Men Volunteers Underscore Metabolically Healthy and Unhealthy Phenotypes

**DOI:** 10.3390/nu12113545

**Published:** 2020-11-19

**Authors:** Antonin Lamaziere, Dominique Rainteau, Pukar KC, Lydie Humbert, Emilie Gauliard, Farid Ichou, Maharajah Ponnaiah, Nadine Bouby, Joe-Elie Salem, Jean-Maurice Mallet, Maryse Guerin, Philippe Lesnik

**Affiliations:** 1INSERM, Saint Antoine Research Center, CRSA, Sorbonne Université, F-75012 Paris, France; antonin.lamaziere@aphp.fr (A.L.); dominique.rainteau@upmc.fr (D.R.); lydie.humbert@upmc.fr (L.H.); Emilie.gauliard@upmc.fr (E.G.); 2Département de Métabolomique Clinique, Hôpital Saint Antoine, AP-HP/Sorbonne Université, F-75012 Paris, France; 3Institut National de la Santé et de la Recherche Médicale (INSERM, UMR_S 1166-ICAN), Sorbonne Université, Research Institute of Cardiovascular Disease, Metabolism and Nutrition, Faculté de Médecine—Hôpital Pitié-Salpêtrière, F-75013 Paris, France; pukar.kc@inserm.fr (P.K.); joe-elie.salem@aphp.fr (J.-E.S.); maryse.guerin@inserm.fr (M.G.); 4Institute of Cardiometabolism and Nutrition (ICAN), 75013 Paris, France; f.ichou@ican-institute.org (F.I.); m.ponnaiah@ican-institute.org (M.P.); 5Centre de Recherches des Cordeliers (CRC), INSERM, UMRS 1138, F-75006 Paris, France; nadine.bouby@inserm.fr; 6Centre d’Investigation Clinique Paris-Est CIC-1901, Department of Pharmacology, Hôpital de la Pitié-Salpêtrière AP-HP, 73013 Paris, France; 7Département de Chimie, École Normale Supérieure, PSL University, 75005 Paris, France; jean-maurice.mallet@ens.fr; 8Laboratoire des Biomolécules, LBM, Sorbonne Université, CNRS, 75005 Paris, France

**Keywords:** bile acids, postprandial, microbiota, lipoproteins, LDL, HDL, triglycerides, health status, profiling

## Abstract

Bile acids (BAs) regulate dietary lipid hydrolysis and absorption in the proximal intestine. Several studies have highlighted a determinant role of circulating levels and/or metabolism of BAs in the pathogenesis of major cardiometabolic diseases. Whether changes in BA profiles are causative or are consequence of these diseases remains to be determined. Healthy male volunteers (n = 71) underwent a postprandial exploration following consumption of a hypercaloric high fat typical Western meal providing 1200 kcal. We investigated variations of circulating levels of 28 BA species, together with BA synthesis marker 7α-hydroxy-4-cholesten-3-one (C4) over an approximately diurnal 12 h period. Analysis of BA variations during the postprandial time course revealed two major phenotypes with opposite fluctuations, i.e., circulating levels of each individual species of unconjugated BAs were reduced after meal consumption whereas those of tauro- and glyco-conjugated BAs were increased. By an unbiased classification strategy based on absolute postprandial changes in BA species levels, we classified subjects into three distinct clusters; the two extreme clusters being characterized by the smallest absolute changes in either unconjugated-BAs or conjugated-BAs. Finally, we demonstrated that our clustering based on postprandial changes in BA profiles was associated with specific clinical and biochemical features, including postprandial triglyceride levels, BMI or waist circumference. Altogether, our study reveals that postprandial profiles/patterns of BAs in response to a hypercaloric high fat challenge is associated with healthy or unhealthy metabolic phenotypes that may help in the early identification of subjects at risk of developing metabolic disorders.

## 1. Introduction

Bile acids (BAs), the primary catabolic products of cholesterol, are synthesized in the liver and generate cholic acid (CA) and chenodeoxycholic (CDCA) in humans. These primary BAs are then amidated on the side chain with either glycine or taurine (GCA, GCDA, TCA, TCDCA) and eventually sulfonated or glucuronidated in the steroid backbone [1,2], then conjugated-BAs (conj-BAs) are released and stored in the gallbladder. After a meal, these conj-BAs are secreted into the intestine to facilitate the emulsification of dietary lipids in the intestinal lumen [1]. Due to their amphiphilic properties, conj-BAs are found adsorbed at the surface of oil-in-water emulsions, where they regulate the lipolytic activity of the pancreatic lipase-colipase complex. Conj-BAs are also found in the form of mixed micelles with other bile lipids (cholesterol, phosphatidylcholine, fat-soluble vitamins) and lipolysis products (free fatty acids, 2-monoglycerides). The micellar co-solubilization of lipolysis products by BAs is required for removing these products from the lipid–water interface and for ensuring their diffusion in the aqueous milieu of the gut towards the enterocytes. In the absence of bile secretion, fat absorption is impaired [3,4]. In the ileum, a highly efficient transporter system allows active reabsorption of conj-BAs back to the liver via mesenteric and portal veins [4].

Further complexity in BA metabolism is the modification of BA structure by intestinal bacteria [5]. Microbial enzymes such as bile salt hydrolases (BSH) deconjugate conj-BAs, bacterial 7a-dehydroxylases, and 7b-dehydroxylases convert the primary CA and CDCA to the secondary BAs deoxycholic acid (DCA) and lithocholic acid (LCA), respectively. Bacteria isomerize the 7a-hydroxyl group in CDCA to the 7b-hydroxyl group in ursodeoxycholic acid (UDCA) and bacterial sulfatases hydrolyze sulfated-BAs of their ester linkage [6,7], ultimately leading to the presence of a large repertoire of secondary BAs. In the colon, unconjugated-BAs (unconj-BAs) produced by microbial metabolism can diffuse passively over the intestinal border and eventually be captured by the liver via transporters of the organic anion transporting polypeptide (OATP) family that can transport unconj-BAs and sulfated-BAs [4]. In the liver, conj-BAs are more efficiently recycled from portal blood at the hepatic basolateral membrane by the high-affinity sodium-dependent taurocholate cotransporting polypeptide (NTCP) than unconj-BAs [1,8,9]. The estimated hepatic fractional uptake of total BAs range from 50 to 90% depending on the bile acid structure [8,9], and is reflected by differences in systemic blood concentration versus portal blood concentration [8]. During the completion of the enterohepatic cycle, unamidated-BAs can be conjugated again in the liver leading to the formation of their glycine or taurine conjugates (GDCA, GLCA, GUDCA, TDCA, TLCA and TUDCA) or the conversion of CDCA and LCA to the 6α-hydroxylated acids hyocholic acid (HCA) and hyodeoxycholic acid (HDCA), respectively [10]. To be exhaustive, peripheral blood concentrations of BAs may also originate from a direct secretion into the lymph and excretion into the circulation through the thoracic duct, albeit that this pathway has been poorly investigated [11].

Significant determinants of the content and composition of the bile acid pool are their efficient enterohepatic recirculation, their host and microbial metabolism, and the homeostatic feedback mechanisms connecting hepatocytes, enterocytes and the luminal microbiota. BAs are hormone-like signaling metabolites that regulate their metabolism and lipid and lipoprotein metabolism, glucose homeostasis, energy expenditure, intestinal motility and bacterial growth, inflammation and carcinogenesis in multiple cell types and tissues through interactions with both host receptors and the microbiota [5]. In recent years, it has been found that BA in the peripheral blood enables the activation of BA signaling outside the enterohepatic system with regulatory functions in the carbohydrate, lipid and energy metabolism by binding to nuclear receptors like the farnesoid X receptor (FXR) and the G-protein coupled receptor TGR5 [12,13,14]. Accordingly, altered bile acid circulation and/or metabolism is now suggested to play a role in the pathogenesis of diabetes/metabolic disorders, cardiovascular diseases or immune-mediated diseases [13,15,16,17]. It remains to determine whether these changes in BA profiles are causative of these diseases or whether the changes in the bile acid pattern and composition are bystanders of these diseases.

In humans, the peripheral venous blood concentrations of total BA at the fasting state is believed to represent the hepatic spillover which constitutes the physiological state in nearly 1% of the whole-body pool [9,18]. Total BA concentrations can vary considerably between individuals at the fasting state [9,19] even more significantly during an oral glucose tolerance test [20,21,22,23,24], or in response to mixed meals [9,19,24,25,26,27] and during the circadian cycle [19,28], albeit with considerable variations for each BA metabolite. Variations in circulating BA concentrations originate from diverse factors including age [29], gender [30], diet composition [31] and individual genetics [24] as well as health status [23,25,32,33]. Nevertheless, there is a paucity of information concerning the potential effect of a hypercaloric fat-rich diet (western pattern diet) on postprandial BA metabolism. Several studies have reported changes in postprandial BA levels after a low-caloric intake [23,25] or after a normal caloric intake [27,34] in various populations. 

In order to better understand the effect of a hypercaloric fat-rich diet challenge on circulating BA levels, we conducted a real-life rhythms postprandial study under highly controlled conditions on 71 healthy men from the “HDL-PP cohort” [35]. We now hypothesize that the postprandial profiles/patterns of BAs in response to an unhealthy diet challenge may reveal the early markers of disease susceptibility and underline the respective involvement of the liver, intestine and microbiota in BA metabolism. Therefore, the present study aimed to quantify the time course/kinetics changes in plasma concentration of 28 BA species (primary, secondary conjugated and unconjugated bile acids) after a western dietary pattern challenge comprehensively. These profile variations were monitored in parallel with the bile acid synthesis marker 7α-hydroxy-4-cholesten-3-one (C4) and parameters of lipid and lipoprotein metabolism, namely triglycerides (TG), low-density lipoprotein (LDL) and high-density lipoprotein (HDL). For the first time, we describe the effects of a typical lipid-rich western meal (1200 kcal; distinguishable by its content of typically consumed solid foodstuffs rather than a liquid formula) on BAs’ quantitative and qualitative features in a normolipidemic male.

## 2. Materials and Methods

### 2.1. Study Population and Design

The study population, which has been previously described [35], was composed of 71 male volunteers aged between 18 and 56 years. This population was representative of a general Caucasian population of healthy men displaying clinical and biochemical fasting parameters within the normal range for their age (Table 1). No volunteers suffered from diabetes, liver, renal or thyroid disease and they were nonsmokers. No subject was under lipid-lowering therapy. Twenty-nine subjects (40.8%) of the study population displayed either 1 of the 5 criteria of the metabolic syndrome that include abdominal obesity characterized by increased waist circumference ≥102 cm for men, fasting triglycerides ≥150 mg/dL, HDL-C levels ≤40 mg/dL for men, elevated blood pressure (systolic blood pressure ≥130 mm Hg and/or diastolic blood pressure ≥85 mm Hg) and fasting blood glucose ≥5.6 mmol/L or met the criteria of prediabetes as defined by the American Diabetes Association [ADA2019] (fasting plasma glucose between 5.6 mmol/L to 6.9 mmol/L and/or (b) impaired glucose tolerance defined as glucose levels between 7.8 mmol/L to 11 mmol/L after a 75 gr oral glucose tolerance test and/or (c) HbA1c plasma levels of 5.7% to 6.4%). Besides, 6 subjects were characterized by the presence of two concomitant components of the metabolic syndrome and 3 subjects older than 45 years exhibited a metabolic syndrome with at least 3 concomitant criteria. 

For each individual, a postprandial exploration was conducted as previously described [36] and as summarized in Figure 1. A baseline fasting blood sample was collected at 8:00 a.m. after 12 h overnight fast (FS: fasting state). A standardized breakfast of low caloric content (300 kcal) was consumed at 8:30 a.m. Subjects consumed the test meal (1200 kcal) at 11:30 a.m. Blood samples were obtained immediately before the consumption of the test meal (T0, 11:30 am) and two (T2; 1:30 p.m.), four (T4; 3:30 p.m.), six (T6, 5:30 p.m.) and eight (T8, 7:30 p.m.) hours after ingestion of the meal. Plasma was separated immediately by low-speed centrifugation (2500 rpm) for 20 min at 4 °C and stored at −80 °C until use. 

### 2.2. Ethics

This clinical protocol (registration number CPP/57-11) was approved by the scientific ethical committee of the Pitié-Salpêtrière hospital, was registered at clinicaltrials.gov (NCT 03109067) and conforms to the principle of the Declaration of Helsinki. Postprandial explorations were carried out at the clinical investigation center of the Pitié-Salpêtrière hospital (CIC1901 Paris-Est). Written informed consent was obtained from each patient.

### 2.3. Extraction of Plasma Bile Acids and Measurements by High Pressure Liquid Chromatography Tandem Mass Spectrometry

**Reagents:** Methanol, acetonitrile and formic acid were of MS grade. Deionized water comes from a Milli-Q Elix system fitted with a LC-PaK and a MilliPak filter at 0.22 μm (Merck Millipore, Guyancourt, France). The following bile acid standards: cholic acid (CA), chenodeoxycholic acid (CDCA), deoxycholic acid (DCA), lithocholic acid (LCA), ursodeoxycholic acid (UDCA), hyodeoxycholic acid (HDCA), hyocholic (HCA), beta-muricholic (βMCA), glycocholic acid (GCA), glycochenodeoxycholic acid (GCDCA), glycodeoxycholic acid (GDCA), glycolithocholic acid (GLCA), glycoursodeoxycholic (GUDCA), taurocholic acid (TCA), taurochenodeoxycholic acid (TCDCA), taurodeoxycholic acid (TDCA), tauroursodeoxycholic acid (TUDCA), taurolithocholic acid (TLCA), taurohyodeoxycholic (THDCA) and ammonium acetate were purchased from Sigma Chemical (St. Louis, MO, USA). LCA3S lithocholic acid 3sulfate was synthetized from LCA in the laboratory. Other sulfated BA were a generous gift from Dr. J. Goto. The internal standards were 23-nor-5β-cholanoic acid-3α,12α diol from Steraloids Inc. (Newport, USA). The 7α-hydroxy-4-cholesten-3-one (C4) and 7α-hydroxy-4-cholesten-3-one-D7 (C4-D7) were obtained from Avanti Polar Lipid. The stock solutions of bile acids, C4 and D7C4 were prepared separately in methanol at the concentration of 10 mmol/L, and the stock solutions were stored at −20 °C. 

**Extraction:** Biological samples (100 µL plasma) were deproteinized by addition of acetonitrile (final concentration 80%, *v*/*v*) containing 25 µg of Nor (23-nor-5β-cholanoic acid-3α,12α diol) and 10 µg of C4-D7(D7, 7α-hydroxy-4-cholesten-3-one). After stirring (1 min vortex mixing) acetonitrile-treated samples were incubated at room temperature for 20 min and centrifuged. The supernatant was evaporated under a nitrogen stream at 50 °C and the residue resuspended in 150 µL methanol; 2 µL was injected.

Bile acids quantification: The separation of BA as a function of polarity was accomplished using an analytical column (Pinnacle II C18; 250 mm × 3.2 mm (L × ID), 5 µm silica particle (Restek, Lisses, France) fitted on an HPLC binary pump Shimadzu Nexera XR system (Shimazu France, Marne la Vallee, France). Column was maintained at 35°C during the analytical run. The mobile phase was composed of a mixture of ammonium acetate (15 mM, pH 5.3) (A) and methanol (B). Gradient settings for eluents A/B (in %) were 70/30 (0 min), 70/30 (1 min), 60/40 ‘3 min), 45/55 (6 min), 40/60 (8 min), 35/65 (12 min, 25/75 (16 min), 5/95 (20 min), 5/95 (22min); 70/30 (23 min) and 70/30 re-equilibration-start (28 min). The flow rate was regulated during the elution run as follows; from 0.3 mL/min (0–20 min) to 0.5 mL/min (20–23 min) and back to 0.3 mL/min (23–28 min). The HPLC was in series with the turbo ion-spray source of the tandem mass spectrometer (Q-trap5500 Sciex, ABSciex, Foster City, CA, USA). Electrospray ionization was performed in negative mode, with nitrogen as the nebulizer gas. The temperature of the evaporation gas was set at 400 °C. The ion-spray voltage, declustering and entrance potentials were set at −4500 V, −60 V and −10 V, respectively. Collision-induced dissociation was achieved in a Q2 collision cell under various voltage potentials, depending on the conjugation table, and MS/MS detection was operated with unit/unit resolution in the multiple-reaction-monitoring (MRM) mode (Appendix A). Data were acquired using Analyst V.1.4.2 software [37] and were quantified using MultiQuant software 3.0 (ABSciex, Foster City, CA, USA). Results are expressed in nmol/L for total BAs and in proportion (%) of total BAs for each specific BA (+-SEM) after calibration of the method, and normalization relative to the internal standard (NOR).

C4 quantification: The separation of C4 and C4-D7 as a function of polarity was accomplished using an analytical column (Pinnacle II C18; 250 mm × 3.2 mm (L × ID), 5 µm silica particle (Restek, Lisses, France) fitted on an HPLC binary pump Nexera XR system (Shimazu France, Marne la Vallee, France). The mobile phase was composed of a mixture of water (0.1% Formic acid) (A) and methanol (0.1% formic acid (B)). Gradient settings for eluents A/B (in %) were 20/80 (0 min), 20/80 (0.3 min), 0/100 (14 min), 0/100 (15 min), 20/80 (15.5 min), 20/80 (18 min), and the flow rate was kept at 0.3 mL/min. The column was maintained at 45°C during the analysis. The HPLC was coupled in series with the turbo ion-spray source of the tandem mass spectrometer (Q-trap5500 Sciex, ABSciex, Foster City, CA, USA). Electrospray ionization was performed in positive mode, with nitrogen as the nebulizer gas. The temperature of the evaporation gas was set at 500 °C. The ion-spray voltage, declustering and entrance potentials were set at −4500 V, 90 V and 10 V, respectively. Collision-induced dissociation was achieved in a Q2 collision cell under various voltage potentials, and MS/MS detection was operated with unit/unit resolution in the multiple-reaction-monitoring (MRM) mode (Appendix A). Data were acquired using Analyst V.1.4.2 software. Upon collection, the LC-MS/MS data were analyzed using MultiQuant software 3.0 (ABSciex, Foster City, CA, USA) with built-in queries or quality control rules allowing us to set compound-specific criteria for flagging outlier results. Flagging criteria included accuracies for standards and quality controls, quantifier ion/qualifier ion ratios, and lower/upper calculated concentration limits. For each calibration curve, the regression line used for quantitation was calculated using least-squares weighting (1/x). C4 results were expressed in ng/mL (+-SEM) [38].

### 2.4. Statistical Analyses 

Variables were tested for normal distribution using the Kolmogorov–Smirnov test. Circulating levels of BAs did not follow a Gaussian frequency distribution. The Friedman test was used to assess changes over postprandial time course. The Dunn’s test was applied for multiple comparisons. Postprandial variations in BA levels were quantified by calculating the absolute change relative to fasting state (FS). For each individual, the maximal absolute postprandial change in BA levels was calculated as the difference between minimal or maximal circulating levels observed from 2 h to 6 h after meal intake and circulating levels determined at FS. Multivariate data analysis were performed by using MetaboAnalyst 4.0 (Xia Lab at McGill University, Montreal, QC, Canada). Absolute changes from FS of individual BA species were analyzed by Ward’s Hierarchical Clustering and visualized with a heatmap. Postprandial variations in triglyceride levels were quantified by calculating the area under the curve (AUC) or incremental AUC (iAUC) determined relative to FS. AUC_0-11h30_ or iAUC_0-11h30_ were calculated by the trapezoidal method for the entire period from FS (8:00 a.m.) to eight hours after meal intake (T8; 7:30 p.m.). Correlations of BA levels with other continuous variables were calculated by using the Spearman rank test. Multiple linear regression analyses were performed using R statistical software computer program version 3.3.1 and were used to determine the impact of clinical parameters on postprandial variations of BA levels assessed by multiple regression analysis including age, BMI and fasting levels of triglycerides, LDL-C, HDL-C and glucose as independent variables and maximal absolute change in maximal absolute postprandial change in bile acid species and C4 as dependent variables. Multiple regression results were expressed as standardized β-coefficient. Results were considered statistically significant at *p* < 0.05.

## 3. Results

### 3.1. Fasting and Postprandial Bile Acid Profiles

A total of 71 human volunteers underwent a whole day experiment involving an overnight fasting period of 12 h followed in the morning by a breakfast of 300 Kcal. Then, after a first postprandial period of 3 h, the volunteers were again challenged with a palatable hypercaloric and lipid-rich diet containing 142 mg of cholesterol and were monitored all afternoon (Figure 1) [35,36,39]. Fasting plasma concentration of BAs showed large variations between individuals (Table 2 and Appendix A). Among the 28 detectable molecular species of BAs, CA-3S, UDCA-3S and TUDCA-3S, which are classically detected in human urine and feces, [40,41] were below the detection limit in plasma (Appendix A). Fifteen BAs were detected in a fraction of the studied population (i.e., LCA, HDCA, HCA, THDCA, TLCA, GHDCA, TUDCA, CDCA-3S, DCA-3S, LCA-3S, UDCA-3S and GUDCA-3S). The inter-individual variabilities expressed as the 95th percentile ratio to the 5th percentile of BAs metabolites detected in the fasted state were maximal for HDCA, LCA, TLCA and TDCA (Table 2), all of which are related to microbial metabolism. In the postprandial state, the maximum inter-individual variabilities were observed for GUDCA-3S, LCA-3S and HCA. Several BA metabolites including GUDCA-3S, LCA, TLCA, DCA-3S, TUDCA, LCA-3S and GLCA were detected in respectively 4%, 11%, 11%, 30%, 30%, 34% and 60% of the whole population (Table 2 and Appendix A). Any of the bile acid species detected in less than 25% of the study population were not included in further analyses.

Hierarchical cluster analysis of BA variations during the kinetic revealed two major phenotypes with opposite fluctuations (Pattern A and B) (Figure 2). Representative kinetics of Pattern A and B are shown on Figure 2B,C, respectively and additional kinetics on Supplemental Appendix A. Hierarchical cluster analyses of major BA classes (Supplemental Appendix A), BA ratios and indexes of hydrophobicity (Supplemental Appendix A) during the time course highlight the general features of the dynamic response to a western-type diet. The fluctuations of unconjugated BAs were inverse to the variations of tauro- and glyco-conjugated BAs. Secondary BAs were rapidly responsive to the diet and peaks at 2 h then decreases rapidly, while primary Bas, including conjugated Bas, return gradually to a basal level within 6 h. By contrast, the patterns of tertiary and sulfated BA responses were smaller and were poorly associated with the return to the fasting state.

Bile acid species are differentially correlated with triglyceride levels according to either fasting or postprandial state (Table 3 and Appendix A). Indeed, at the fasting state, unconjugated BAs, including CA and CDCA, were negatively correlated with TG levels, while these associations were lost during the postprandial period. Conversely, GCA, GUDCA and C4 were positively correlated with TG levels at the fasting state, such an association being more robust in the postprandial state for GCA and GUDCA. Additionally, among the panel of BA species, UDCA displayed a very strong and positive association with TG levels under the postprandial state (*p* < 0.0001), similar to conjugated and unconjugated forms of DCA and TCA, albeit with a lower statistical strength (*p* < 0.004) and an even weaker power with TCDCA (*p* < 0.02). Among the conjugated BAs, in contrast to taurine-conjugated, glycine-conjugated BAs, they correlated positively with TG levels during the postprandial period. Of particular interest, the hydrophobic index of fasting BAs was negatively correlated to fasting TG. This suggests that spillover of a pool of more hydrophilic BAs might be related to concomitant liver-uptake of more hydrophobic BAs during the fasting period, the latter being associated with a higher liver secretion of VLDL-TG, such association would be then lost during the postprandial period.

Analysis of the relationship between BAs levels and clinical and biological parameters of the study population revealed significant correlations of BA levels with age, total cholesterol and LDL-C (Supplemental Appendix A). Equally, unconjugated BAs inversely correlated with BMI, waist circumference or HbAc1. No significant relationship was observed between BA levels and HDL-C or glucose metabolism parameters including, fasting blood glucose, insulin levels and HOMA-IR. By contrast with Steiner’s earlier report [19], we presently did not observe a significant correlation between C4 levels and total cholesterol or LDL-C or glucose metabolism parameters. It is relevant to note that in contrast with this latter study, none of our study population subjects were diabetic and only three subjects exhibited clinical features of metabolic syndrome.

Absolute maximal postprandial change of individual BA species revealed a great heterogeneity with either reduced, decreased or unchanged levels in the post-absorptive state (Figure 3). In particular, postprandial circulating levels of unconjugated BAs (CA, CDCA, DCA and UDCA) were overall reduced, whereas those of conjugated BAs were increased. 

Absolute postprandial variation in the primary BA levels thus reflected the concomitant postprandial increase in GCDCA, TCDCA, and the decrease in CA and CDCA. The secondary BAs contributing to variations in absolute postprandial changes were DCA, and its conjugated forms, which behave in opposite ways, similar to the tertiary bile acids UDCA and GUDCA. Among sulfated BAs, the most significant maximal absolute changes were due to postprandial increase concentration of GLCA-3S and TLCA-3S.

### 3.2. Classification of the Study Population Based on the Absolute Maximal Postprandial Change in Bile Acid Levels

As shown in Table 4, multiple regression analyses for the association of maximal absolute postprandial change in TCA, TCDCA, GCDCA, GDCA and conjugated BAs positively correlated with fasting LDL-C levels. A similar trend was observed for both GCA and GUDCA; however, such relationships did not reach statistical significance (*p* < 0.01). In addition, multiple regression analyses for the association of absolute maximal postprandial variations in TCA, UDCA, sulfo-conjugated and secondary BA levels negatively correlated with fasting HDL-C levels. It is worth noting that similar non-significant trends were observed for both DCA (*p* < 0.01), TDCA (*p* = 0.055) and GLCA-3S (*p* = 0.072). In contrast, maximal absolute postprandial variations in GUDCA levels were positively correlated with fasting HDL-C levels and correlated negatively with age. Finally, no significant associations between the study population’s clinical parameters, including BMI or fasting glucose levels, and absolute maximal postprandial variations in BA levels or any individual BA species, were observed.

Given the individual heterogeneity of the absolute maximal postprandial changes of circulating BA species levels relative to the fasting state, we categorized the postprandial BA responses for each individual. Accordingly, we carried out an unsupervised clustering approach using an agglomerative hierarchical cluster model built applying Ward’s linkage method to a matrix of Euclidean distances. By combining absolute postprandial changes of circulating levels of BA species of the 71 individuals, we identified three clusters referred to C0, C1 and C2, accounting for 57, 9 and 5 subjects, respectively (Figure 4A). As shown in Figure 4B, in the principal component analysis (PCA) scores plot the three independent clusters were well separated from each other, thus indicating a distinct degree of postprandial change in BA levels between the subgroups of subjects. The first two components (PC1 and PC2) explained more than 65% of the total variance in absolute maximal changes of circulating levels of BA species relative to the fasting state. Moreover, PC1 reflected absolute changes in conjugated BAs, whereas PC2 indicated those of unconjugated BAs (Figure 4C). Individuals belonging to cluster C1 were characterized by the smallest absolute changes in unconjug-BAs as a result of low fasting levels of unconj-BAs; those individuals equally displayed the highest absolute changes in conjug-BAs (Figure 4D and Appendix A). By contrast, subjects with pattern C2 were primarily characterized by low absolute changes in conjug-BAs (Figure 4D and Appendix A). The cluster C0, representing 80% of the cohort, displayed intermediate ranges in postprandial responses for unconjug-BAs - and conj-BAs (Figure 4D and Appendix A). Figure 5 highlights the postprandial-TG (PP-TG) responses associated with clusters C0, C1 and C2. The smallest PP-TG-response associated with cluster C2 and the largest with cluster C1. Conversely, cluster C2 was associated with poor PP-TG response (lowest AUC TG and iAUC TG) as compared with C0 and C1. Cluster C1 was associated with elevated BMI and waist circumference index.

Finally, in an attempt to analyze how subjects with clinical features of the metabolic syndrome (MS) were distributed among clusters, we determined the relative proportion of subjects from clusters C0, C1 and C2 exhibiting MS criteria. For that purpose, the study population was categorized as follows: MS0, patients without out any MS component (n = 45); MS1, patients with 1 MS component (n = 16); MS2, patients with 2 MS component (n = 7); MS3, patients with 3 MS components (n = 3). Due to the low number of patients displaying three criteria of MS, we merged MS2 and MS3 in a single group, MS > 2 (n = 10). Interestingly, the proportion of subjects exhibiting at least two criteria of MS was increased in Cluster 1 (22%) as compared to Cluster 0 (14%). In contrast, none of the subjects from Cluster 2 exhibited two or more concomitant criteria of MS criteria. More strikingly, as shown in Figure 6, unconjugated BA profiles of subjects from the MS > 2 group displayed a similar behavior as those observed for subjects from Cluster 1, thus indicating that Cluster 1 represents an unhealthy BA profile. Equally, conjugated BA profiles of subjects from MS0 was equivalent to those observed for subjects from Cluster 2, thus indicating a healthier feature of such latter BA profiles.

## 4. Discussion

Health is defined as the capacity of an organism to adapt to challenges. In this study, we tested to what extent comprehensively phenotypes of bile acid species in healthy individuals reveal differences in metabolic response to a hypercaloric meal. BA analysis was performed in 71 individuals by profiling 28 BA species over approximately 12 h. According to postprandial measurements, unsupervised BAs molecular species’ clustering discriminated both kinetics-based and subjects-based patterns of variation.

Although previous studies have demonstrated postprandial variations of BA concentrations with standard meals and high glucose challenges [24], the influence of a hypercaloric-lipid-rich diet has not been much investigated. Since most people consume fat-containing meals at regular intervals (4–5 h) as well as fat-containing snacks, we believe that both the tested meals, representative of a typical western meal, and the study design used in the present study can better mimic the real-life postprandial state and challenge the “metabolic flexibility” of healthy people as well as identify metabolically compromised patients [42]. In this context, the lowest concentration of peripheral total BAs at the fasting state was 0.8 µM, contrasting with the maximal postprandial concentration peaking at 17.6 µM (Table 2), thus indicating the great variability and magnitude in total BA concentrations in healthy individuals. This difference of 22 times between the minimum and maximum concentrations are compatible with earlier studies showing that among endogenous metabolites, BAs display the highest inter-individual variability [19,34,43]. More specifically, we also show a maximal variability of 60-fold and 27-fold for conj-BAs and unconj-BAs concentrations, respectively, irrespective of the individuals, albeit in an opposite dynamic. Such a magnitude could be compatible with the pharmacology of the farnesoid X receptor (FXR, NR1H4), and the cell surface G-protein-coupled receptors TGR5 (GPBAR1), for which median effective concentrations of BA species (EC50) were reported and ranged from 0.3 to 10 μM [44,45,46,47]. By contrast, in this healthy population, LCA that were present at low or undetectable circulating levels in both fasting and postprandial states might be unlikely signal through other BA receptors such as the vitamin D receptor (VDR, NR1H1) or the pregnane X receptor (PXR, NR1H2) due to EC50 ranging from 8 to 10 µM [48,49]. After the meal, concentrations of unconjugated BA species decrease, reflecting a fall in circulating levels of all unconjugated BAs species (CA, CDCA, DCA, UDCA). Conversely, all taurine- and glycine-conjugated BA concentrations rise during the postprandial phase, as well as those of two sulfated forms (T-LCA-3S, G-LCA-3S). Such opposite behavior is consistent with a similar scaled study performed following an oral tolerance test containing exclusively lipids (84.2 g) [50] that identified unconjugated BAs (i.e., CDCA, LCA and trends for CA, UDCA and DCA) not contributing to the overall rise in postprandial total BAs commonly observed in numerous studies (review [34]). To note, Sonne et al. did not observe such a postprandial decrease in unconjugated BAs (total unconjugated, CA, CDCA, DCA and UDCA) [23] in a study performed in 15 healthy control subjects challenged with a high fat-containing meal (40 g). Equally, nor did studies of Hauelser et al. [25] conducted on 11 non-obese control and obese (n = 32) subjects challenged with a 509 kcal meal containing 28% of fat or those of Ewang-Emukowhate et al. [51] on 10 healthy males challenged with a 400 kcal meal containing 28.5g of fat report a specific postprandial decrease in unconjugated BAs. It is likely that apparent conflicting results between these latter reports and our present study primarily result from the distinct study population and the fat content of the meal used. However, factors influencing plasma bile acid dynamics may involve numerous mechanisms which were initially documented in the early works of Hofmann et al. [52] and LaRusso et al. [53] This would need to be investigated more specifically in future studies (e.g., BA synthesis capacity, hepatic spillover and uptake, intestinal uptake, colon transit, microbial transformations and others [8]) in the light of the current models of integration [54,55,56,57]. Interestingly, some potential determinants were explored in kinetic studies in a pig trans-organ flux model, in which Eggink et al. [8] determined that the timing and magnitude of the postprandial response exhibited large interindividual variabilities for BAs compared to glucose and insulin responses. Moreover, they showed that the liver selectively extracted most BAs with high TGR5 affinity, with a more efficient clearance for conj-BAs than for unconjugated forms. Additionally, Steiner et al. [19], in a princeps study, evaluated the diurnal variation of 15 BAs and their association with the biosynthetic precursor 7a-hydroxy-4-cholesten-3-one (C4) in four healthy subjects. Despite the large interindividual variability, the authors notably showed dissociation in the diurnal rhythm of the BA biosynthesis marker C4 with the peaking of postprandial conj-BA, which was confirmed in later studies [58], as well as in the present study (disconnected from the postprandial response; Supplemental Appendix A). By contrast, variations of unconj-BAs were mostly reported to be nocturnal and early in the morning [19,58] with large interindividual variability presumably due to altered microbial activity on BAs (deconjugation, dihydroxylation and epimerization) being associated with the circadian remodeling of the microbiota [59]. Therefore, we can hypothesize that differences between Cluster 1 (that display the lowest levels of unconjug-BAs at the fasting state early in the morning) and Cluster 0 or Cluster 2 are potentially related to gut microbial related activities in this subgroup of individuals. Such a hypothesis was previously explored by Fiamoncini et al. [24] albeit with a small impact of the microbiota that was evaluated at the phylogenetic levels. Future studies, through metagenomic analyses of bile salt hydrolase (BSH), hydroxysteroid dehydrogenases (HSDHs), 7α/β-dehydroxylase, 7α/β-epimerase, 3α/β-epimerase and 5β/α- epimerase would undoubtedly help to elucidate these relationships. Additionally, such a relationship may involve unconj-BA influences on circadian rhythms, as suggested by the recent work of Govindarajan et al. [60] Genetic determinants of the postprandial response were also screened in the study of Fiamoncini et al. [24] They, indeed, identified a variant in the SLCO1A2 gene encoding the small intestinal BA transporter organic anion-transporting polypeptide-1A2 (OATP1A2), that was associated with a delay in the postprandial rise of glycine-conj-BA and taurine-conj-BA but not of non-conj-BAs, thus corroborating a slower intestinal uptake of conj-BA [24]. The variant was present at a frequency of 12% in this British population, which was close to the frequency (7%) of the healthy subjects belonging to Cluster 2 and displaying a similar delay and a lower postprandial response as compared to subjects from Cluster 0 or Cluster 1 (Figure 4 and Appendix A). 

In this study, we observed a negative correlation between CA or CDCA levels and circulating levels of TG at the fasting state. In contrast, such relationships were lost in the postprandial state during which positive correlations took over between postprandial taurine- or glycine-conjugated forms of CA and CDCA and postprandial-TG. Seminal studies from the 70s suggest that BAs may lower circulating levels of TG by repressing both hepatic TG production/secretion and stimulating plasma TG clearance. This reciprocal relationship is mostly based on the impact of the pharmacological action of BA-binding resins that induce hepatic production of TG-rich lipoprotein VLDL [61,62], whereas conversely, treatment of cholesterol gallstones with the hydrophobic CDCA reduced hypertriglyceridemia [63,64], likely through hepatic FXR activation [65]. Thus, the presently observed inverse correlations between TG and CDCA or CA at the fasting state are entirely consistent with the very effective endogenous agonist FXR activity of CDCA and CA reported earlier [44,63]. The coupling between BAs and TG levels may equally involve alternative mechanisms and contribution of nuclear receptors or transcriptional factors (i.e., short heterodimer partner (SHP, NR0B2) hepatocyte nuclear factor 4α (HNF4α), liver X receptors (LXRs) or peroxisome proliferator-activated receptors (PPAR)α) (review [66]). Microbial-mediated mechanisms may also contribute, although the causal mechanisms are unclear. In this context, it is noteworthy that gut microbial diversity was associated with both VLDL particles and circulating TG levels, independent of age, sex, BMI, and medication in individuals from the Rotterdam Study and the LifeLines-DEEP cohorts [67]. 

The positive correlation of conjugated forms of CA and CDCA with TG levels during the postprandial state might reflect their emulsification capacity of dietary lipids, allowing efficient lipid hydrolysis and absorption in the proximal intestine. Such a hypothesis was initially proposed for an explanation of impaired absorption of dietary TG in Cyp8b1 deficient mice (12α-hydroxylase) that lack 12α-hydroxylated-BAs (12-OH-BAs) (CA and DCA, and their conjugated forms) [68,69]. Indeed, Cyp8b1 deficiency in mice did not impact total BA levels, thus indicating that the impairment in lipid absorption was related to the lack of 12-OH-BAs and not to the inefficacy of the remaining non-12-OH-BAs to promote lipid absorption. In the present human study, kinetics and magnitude of postprandial 12-OH-BAs, non-12-OH-BAs and TG responses were matching for the three clusters C0, C1 and C2 (Appendix A), we, therefore, conclude that such a mechanism might not be equally relevant in healthy subjects, consistent with an earlier report showing that CYP8B genetic variants were not associated with distinct postprandial kinetics of plasma BA species in 72 healthy subjects.

## 5. Conclusions

Maintaining the precise balance between qualitative (species) and quantitative (amounts) features, as well as preventing the accumulation of excessive BAs in the whole body, are of critical importance under physiological or pathophysiological conditions. In this context, postprandial metabolic response may help to monitor individual resilience to a hypercaloric-lipid-rich diet. The present comprehensive postprandial response analysis of individual bile acid species allowed us to identify two kinetics-based and three subjects-based patterns of bile acid variation. Such patterns were associated with specific clinical and biochemical characteristics such as reduced postprandial TG response, elevated waist circumference, or BMI, thus suggesting that postprandial BAs’ responses might reflect either healthy or unhealthy metabolic phenotypes. Our findings thus suggest that postprandial BA profiling can predict health status. Future studies are required to evaluate whether postprandial changes in BA species may serve as a diagnostic tool for the early identification of subjects with metabolic anomalies prior to appearance of clinical evidence of metabolic disorders.

## Figures and Tables

**Figure 1 nutrients-12-03545-f001:**
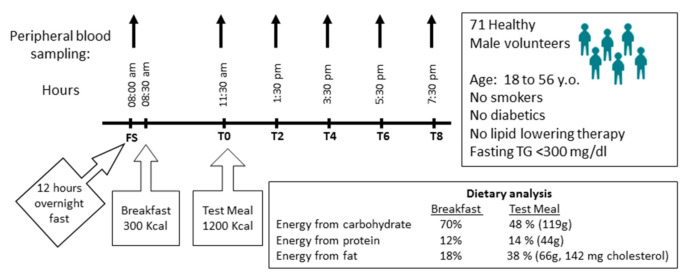
Study design.

**Figure 2 nutrients-12-03545-f002:**
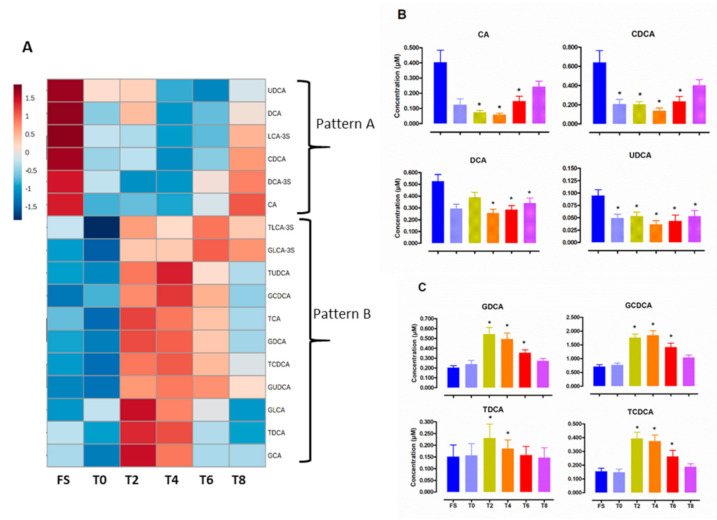
Postprandial levels of individual BA species during postprandial exploration: FS for overnight fasting samples, T0, T2, T4, T6, T8 for before and 2 h, 4 h, 6 h and 8 h after consumption of a hypercaloric high fat test meal. (**A**) Hierarchical cluster analysis of plasma concentrations of BA species according to postprandial time course: Mean circulating levels of individual species of unconjugated BAs (**B**) and of conjugated BAs (**C**) during postprandial exploration. Values are mean ± SEM. * *p* < 0.05 versus overnight fasting sample.

**Figure 3 nutrients-12-03545-f003:**
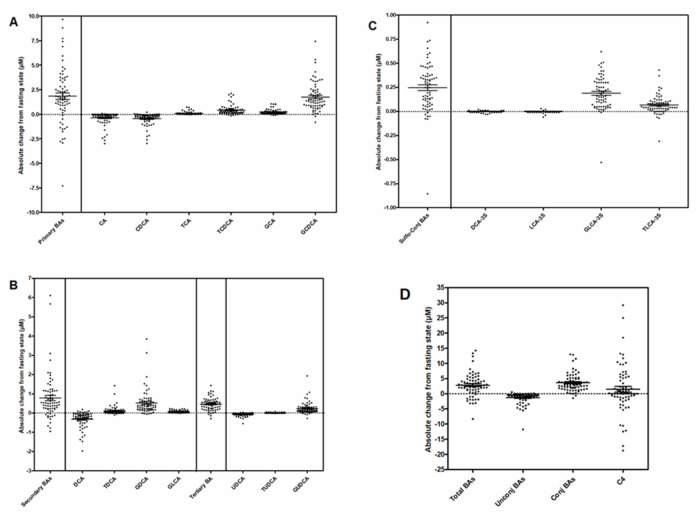
Interindividual variability of maximal absolute postprandial change from fasting sate in circulating BA levels. (**A**) Primary BAs. (**B**) Secondary and Tertiary BAs. (**C**) Sulfo-conjugated BAs. (**D**) Total BAs, unconjugated, conjugated BAs, and BA synthesis marker 7α-hydroxy-4-cholesten-3-one, C4.

**Figure 4 nutrients-12-03545-f004:**
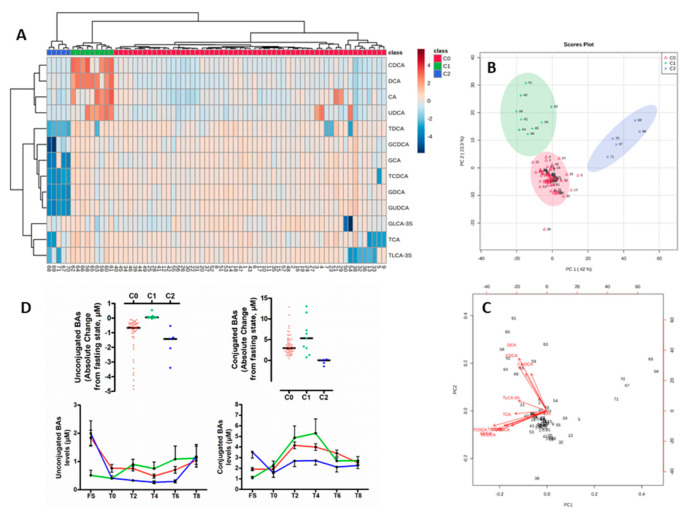
(**A**) Hierarchical cluster analysis of subjects based on their corresponding maximal absolute postprandial change from fasting state in circulating levels of individual BA species. Three clusters were identified and referred to C0 (n = 57, red), C1 (n = 9, green) and C2 (n = 5, blue). (**B**) PCA score. (**C**) PCA biplot. (**D**) Distribution of maximal absolute postprandial change from fasting state of unconjugated and conjugated. Black bars indicate median (upper panel) BAs according to clusters. Circulating levels of unconjugated and conjugated BAs according to postprandial time course in subjects from cluster C0, C1 and C2. Values are mean ± SEM. (bottom panel).

**Figure 5 nutrients-12-03545-f005:**
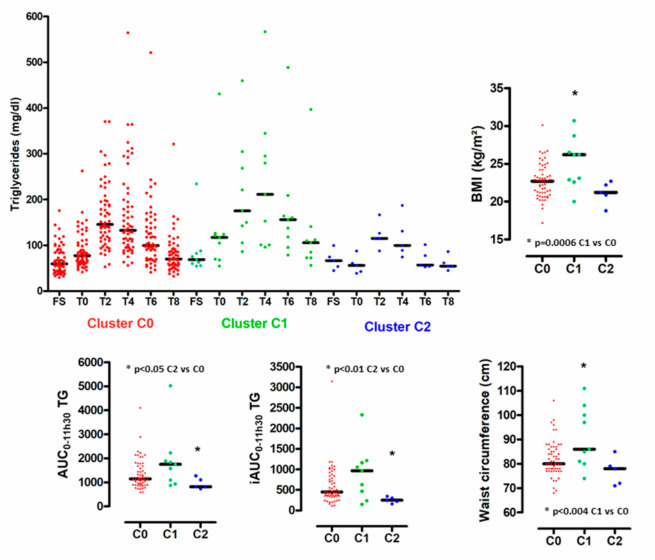
Biological signatures associated with postprandial bile acid profiles. Distribution of plasma triglyceride levels according to postprandial time course in subjects from cluster C0, C1 and C2. (left upper panel). Distribution of AUC_0-11h30_–TG, iAUC_0-11h30_-TG BMI and waist circumference in subjects from cluster C0, C1 and C2. Black bars indicate median.

**Figure 6 nutrients-12-03545-f006:**
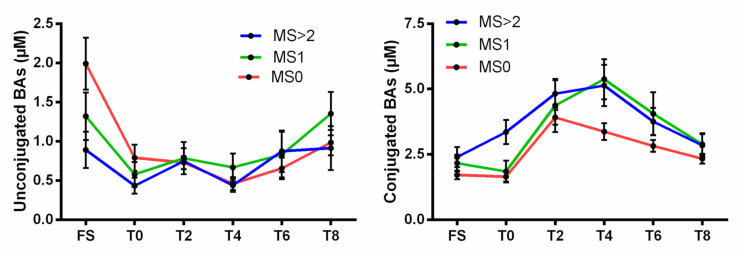
Circulating levels of unconjugated and conjugated BAs according to postprandial time course in subjects stratified according the number of co-existing criteria of the metabolic syndrome (MS). MS0: patients without out any MS component (n = 45); MS1: patients with 1 MS component (n = 16); MS > 2: patients with at least 2 MS components (n = 10). Values are mean ± SEM.

**Table 1 nutrients-12-03545-t001:** Clinical and biological characteristic of the study population.

	Male Healthy Volunteers (n = 71)				
Variables	Median	Q1	Q3	Mini	Maxi
Age, year ≥45 years, n (%)	24.0	22.0	36.0	18.0	56.0
	13 (18.3)				
BMI, kg/m² >30 kg/m², n (%)	22.7	21.2	24.5	17.2	30.7
	2 (2.8)				
Waist circumference (cm) ≥102 cm, n (%)	81.0	78.0	87.0	68.0	111.0
	3 (4.2)				
Systolic blood pressure, mmHg ≥130 mmHg, n (%)	118.0	114.0	126.0	101.0	143.0
	11 (15.5)				
Diastolic blood pressure, mmHg ≥85 mmHg, n (%)	75.0	71.0	80.0	59.0	98.0
	15 (21.1)				
Fasting blood glucose, mmol/L ≥5.6 mmol/L, n (%)	4.60	4.40	5.00	3.30	6.10
	2 (2.8)				
Hba1c, % 5.7% to 6.4%, n (%)	5.30	5.10	5.50	4.40	6.30
	9 (12.7)				
Insulin, mU/L	3.44	1.41	6.24	0.28	11.3
HOMA-IR	0.69	0.27	1.19	0.05	2.40
Creatinine, µmol/L	88.0	80.0	96.0	72.0	108.0
Creatinine Clearance, mL/min	112.2	95.4	127.8	73.2	171.1
ASAT, UI/L	25.0	22.0	29.0	16.0	37.0
ALAT, UI/L	20.0	16.0	31.0	11.0	54.0
TSH, mUI/L	1.58	1.20	2.12	0.58	3.67
CRPus, mg/L	0.50	0.11	1.39	0.01	6.76
Fasting Lipid phenotyping					
Total cholesterol, g/L >2.5 g/L, n (%)	1.73	1.43	1.95	1.01	2.68
	1 (1.4)				
LDL-Cholesterol, g/L >1.6 g/L, n (%)	1.03	0.81	1.20	0.49	1.71
	2 (2.8)				
HDL-Cholesterol, g/L ≤0.40 g/L, n (%)	0.48	0.41	0.56	0.21	0.77
	14 (19.7)				
Triglycerides, g/L >1.5 g/L, n (%)	0.63	0.44	0.85	0.30	2.34
	2 (2.8)				
Postprandial Triglycerides, g/L					
Before test meal, T0: 11:30 a.m.	0.77	0.62	1.10	0.39	4.31
2 h after test meal, T2: 1:30 p.m.	1.46	1.15	1.98	0.53	4.60
4 h after test meal, T4: 3:30 p.m.	1.33	0.96	2.11	5.60	5.67
6 h after test meal, T6: 5:30 p.m.	1.01	0.70	1.56	0.41	5.21
8 h after test meal, T8: 7:30 p.m.	0.71	0.56	0.93	0.32	3.97

BMI: body mass index; ASAT: aspartate aminotransferase; ALAT: alanine aminotransferase; TSH: thyroid-stimulating hormone; LDL: low density lipoprotein; HDL: high density lipoprotein.

**Table 2 nutrients-12-03545-t002:** Descriptive statistics of circulating bile acids levels in fasting and postprandial states.

	Fasting State (n = 71)						Postprandial State (n = 71)						Fold Change from Fasting State
BAs (µM)	Median	Q1	Q3	Mini	Maxi	Interindividual Variability	Median	Q1	Q3	Mini	Maxi	Interindividual Variability	
**Primary BAs**	**1.451**	**0.859**	**2.497**	**0.289**	**10.84**	**1.6**	**3.185**	**2.581**	**4.486**	**0.694**	**11.95**	**1.3**	**2.2**
Unconj													
CA	0.110	0.040	0.380	<	3.050	2.3	0.020	0.010	0.040	0.010	0.300	1.9	−5.5
CDCA	0.310	0.150	0.640	0.010	3.050	1.8	0.050	0.030	0.090	0.010	1.340	2.7	−6.2
Taurine-Conj													
TCA	0.020	0.010	0.050	<	0.890	3.4	0.080	0.040	0.140	<	1.060	1.9	4.0
TCDCA	0.090	0.040	0.200	<	1.030	1.8	0.380	0.190	0.700	0.050	3.020	1.6	4.2
Glycine-Conj													
GCA	0.070	0.040	0.150	0.010	0.680	1.6	0.270	0.150	0.370	0.060	1.310	1.5	3.9
GCDCA	0.500	0.360	0.930	0.070	2.790	1.4	2.020	1.460	3.160	0.520	9.050	1.3	4.0
**Secondary BAs**	**1.095**	**0.662**	**1.580**	**0.085**	**5.653**	**1.4**	**1.617**	**1.247**	**2.580**	**0.190**	**8.026**	**1.4**	**1.5**
UnConj													
DCA	0.420	0.180	0.660	0.020	2.430	1.5	0.180	0.080	0.280	0.010	0.860	1.5	−2.3
LCA	<	<	<	<	0.010	5.0	0.010	0.010	0.020	<	0.120	1.8	na
HDCA	<	<	<	<	0.110	11.5	<	<	<	<	0.210	na	na
HCA	<	<	<	<	0.090	4.7	<	<	<	<	0.100	4.3	na
Taurine-Conj													
TDCA	0.090	0.060	0.120	<	3.600	4.9	0.160	0.100	0.250	<	4.020	2.6	1.8
THDCA	<	<	<	<	0.060	na	<	<	<	<	0.030	na	na
TLCA	<	<	<	<	0.010	5.0	0.010	0.010	0.020	<	0.120	1.8	na
Glycine-Conj													
GDCA	0.190	0.070	0.280	<	1.000	1.5	0.570	0.360	0.910	0.040	3.910	1.5	3
GHDCA	<	<	<	<		na	<	<	<	<	<	na	na
GLCA	0.010	<	0.010	<	0.050	1.7	0.070	0.040	0.120	<	0.240	1.4	7
**Tertiary BAs**	**0.488**	**0.304**	**0.650**	**0.025**	**1.617**	**1.3**	**0.958**	**0.730**	**1.130**	**0.288**	**2.286**	**1.2**	**1.96**
Unconj													
UDCA	0.060	0.030	0.110	<	0.550	1.7	0.020	0.010	0.040	<	0.380	2.2	−3
Taurine-Conj													
TUDCA	<	<	0.010	<	0.060	2.9	0.010	<	0.020	<	0.120	1.7	na
Glycine-Conj													
GUDCA	0.090	0.060	0.180	<	1.420	1.9	0.270	0.160	0.440	0.030	2.100	1.6	3.0
**Sulfo-Conjugated**	**0.300**	**0.194**	**0.470**	**<**	**1.543**	**1.4**	**0.571**	**0.416**	**0.783**	**0.017**	**1.530**	**1.3**	**1.9**
CA-3S	<	<	<	<	<	na	<	<	<	<	<	na	na
CDCA-3S	<	<	<	<	0.020	3.5	<	<	<	<	0.010	na	na
DCA-3S	<	<	0.010	<	0.040	2.4	<	<	0.010	<	0.040	2.9	na
LCA-3S	<	<	0.010	<	0.070	3.0	<	<	<	<	0.040	4.3	na
TLCA-3S	0.100	0.050	0.170	<	0.550	1.5	0.150	0.100	0.260	<	0.800	1.4	1.5
GLCA-3S	0.190	0.110	0.270	<	1.100	1.5	0.390	0.240	0.550	<	1.150	1.3	2.1
UDCA-3S	<	<	<	<	<	na	<	<	<	<	<	na	na
TUDCA-3S	<	<	<	<	<	na	<	<	<	<	<	na	na
GUDCA-3S	<	<	<	<	0.020	na	<	<	<	<	0.020	10.6	na
**Total BAs**	**2.823**	**1.974**	**4.862**	**0.799**	**13.55**	**1.4**	**5.696**	**4.355**	**7.591**	**2.104**	**17.59**	**1.2**	**2.0**
Unconj BAs	0.962	0.572	1.882	0.099	11.97	1.7	0.309	0.173	0.418	0.043	2.660	1.6	−3.2
Conj BAs	1.569	1.081	2.527	0.264	5.292	1.3	4.606	3.526	6.778	1.780	15.620	1.3	2.9
**C4**	**2.972**	**0.036**	**8.746**	**<**	**33.35**	**1.8**	**5.645**	**1.045**	**10.800**	**<**	**31.670**	**1.6**	**1.9**

Interindividual variability expressed as the ratio of 95th percentile/5th percentile; Conj: conjugated; Unconj: unconjugated; Taurine-conj: Taurine-conjugated; Glycine-Conj: Glycine-conjugated. na indicates: not applicable.

**Table 3 nutrients-12-03545-t003:** Relationship between circulating levels of bile acid species and triglycerides in both fasting and postprandial states.

	Fasting State		Postprandial State	
	Spearman r	*p*-Value	Spearman r	*p*-Value
**Primary Bile Acids**	−0.002	**0.97**	**0.184**	**0.007**
CA	**−0.211**	0.002	−0.046	0.50
CDCA	**−0.163**	0.01	0.079	0.25
TCA	0.127	0.09	**0.198**	**0.004**
TCDCA	0.061	0.37	**0.152**	**0.02**
GCA	**0.166**	**0.01**	**0.199**	**0.003**
GCDCA	0.124	0.07	**0.169**	**0.01**
**Secondary Bile Acids**	−0.045	0.50	**0.139**	**0.04**
DCA	−0.112	0.10	**0.197**	**0.004**
TDCA	0.103	0.38	**0.198**	**0.004**
GDCA	0.093	0.17	**0.184**	**0.007**
**Tertiary Bile Acids**	0.014	0.83	0.102	0.13
UDCA	−0.056	0.43	**0.302**	**0.0001**
GUDCA	**0.174**	**0.01**	**0.193**	**0.004**
**Sulfo−Conjugated**	−0.086	0.21	−0.015	**0.82**
TLCA−3S	−0.112	0.10	−0.001	0.98
GLCA−3S	−0.035	0.60	0.038	0.57
**Total BAs**	−0.021	0.75	**0.218**	**0.001**
Unconjugated BAs	**−0.163**	0.01	**0.156**	**0.02**
Conjugated BAs	0.095	0.16	**0.178**	**0.009**
**C4**	**0.205**	**0.01**	**0.167**	**0.03**
Primary BAs/Secondary BAs	0.025	0.71	0.046	0.49
Glycoconjugated−BAs	0.127	0.06	**0.201**	**0.003**
Tauroconjugated−BAs	0.024	0.72	0.119	0.08
Hydrophobic Index	**−0.142**	**0.03**	0.083	0.22

Samples collected after an overnight fast (FS), before (T0) and after 8 h (T8) after test meal intake was used to assess the relationship between bile acids and triglyceride levels in the fasting state (n = 213). Samples collected 2 h (T2), 4 h (T4) and 6 h (T6) after test meal intake was used to assess the relationship between bile acids and triglyceride levels in the postprandial state (n = 213). Statistically significant correlations are printed in boldface.

**Table 4 nutrients-12-03545-t004:** Multiple regression analyses for the association of maximal absolute postprandial change in bile acid species with fasting lipid levels or selected clinical parameters.

	R²	TG	LDL-C	HDL-C	Age	BMI	Glucose
**Primary Bile Acids**	0.125	0.074	**0.325**	−0.134	−0.050	−0.170	−0.065
CA	0.054	0.096	0.015	−0.152	0.120	−0.059	0.040
CDCA	0.133	0.198	0.200	−0.103	0.127	−0.038	−0.159
TCA	0.153	−0.044	**0.344**	**−0.257**	−0.015	−0.129	−0.145
TCDCA	0.116	−0.061	**0.368**	−0.098	−0.044	−0.110	0.004
GCA	0.098	0.079	0.232	−0.116	0.014	−0.251	0.094
GCDCA	0.114	−0.029	**0.307**	0.052	−0.294	−0.061	0.098
**Secondary Bile Acids**	0.168	−0.036	0.256	**−0.306**	0.093	−0.250	−0.128
DCA	0.141	0.188	−0.080	−0.199	0.256	−0.258	−0.030
TDCA	0.122	−0.132	0.205	−0.235	−0.036	−0.249	−0.025
GDCA	0.136	−0.075	**0.319**	−0.190	−0.037	−0.257	−0.067
**Tertiary Bile Acids**	0.104	0.074	**0.277**	0.161	−0.177	0.117	0.005
UDCA	0.205	0.125	0.153	**−0.277**	0.149	−0.154	0.176
GUDCA	0.216	0.043	0.246	**0.328**	**−0.347**	0.128	0.183
**Sulfo-Conjugated**	0.153	0.078	0.080	**−0.299**	0.218	−0.135	−0.161
TLCA-3S	0.129	0.047	0.133	−0.203	0.269	−0.152	−0.121
GLCA-3S	0.027	0.048	0.031	−0.222	0.178	−0.041	−0.192
**Total BAs**	0.138	0.046	**0.346**	−0.171	−0.037	−0.189	0.027
Unconjugated BAs	0.142	0.176	0.093	−0.219	0.188	−0.181	0.041
Conjugated BAs	0.105	−0.037	**0.347**	−0.043	−0.130	−0.179	0.013
**C4**	0.103	−0.167	0.030	0.027	−0.222	0.321	−0.029

Values indicated standardized β regression coefficient. TG: fasting triglycerides; LDL-C: LDLC-Cholesterol; HDL-C: HDL-Cholesterol; BMI: Body Mass Index. Statistically significant and independent associations are highlighted in bold face while trends (*p* < 0.01) are underlined.

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
