# Peer review of "Distinct Postprandial Bile Acids Responses to a High-Calorie Diet in Men Volunteers Underscore Metabolically Healthy and Unhealthy Phenotypes"

_nutrients, 2020, doi:10.3390/nu12113545_

Round 1

Reviewer 1 Report

Authors present data on the relation between the circulating plasma concentration of various bile acids and the consumption of a hypercaloric high fat meal. They also try to highlight a possible correlation with healthy and unhealthy phenotypes. The work is interesting, but in my opinion substantial improvements are needed.

INTRODUCTION

LINE 83 (and in other lines of the text): "size and composition" should be replaced with "content and composition"

LINE 97 (and in other lines of the text): "in the fasting" should be replaced with "at fasting"

 MATERIALS AND METHODS

  • Please describe how samples were collected, in particular report the type of tubes used
  • Please report more details on the analytical method and the extraction procedure used. No information on bile acid extraction are given, nor about standards and solvents used. The instrumental method reported is incomplete. HPLC-MS parameters are not well defined, e.g. mobile phase gradient, injection volume, etc. I suggest including in a Table all MRM parameters. You should report parent ion, fragments and dissociation energy used. It should be grateful if you include in the table the molecular formula and structure for the analytes, divided by class, i.e. uncoji-bas, conji-bas, primary and secondary. Please report also some information about method validation, i.e. limit of detection and quantification for each analytes, recovery, calibration curve information.
  • Why did you not try different collision energies for analytes that were not fragmented (line 180)?
  • Which are the procedures used for the determination of other markers, such as C4?

RESULTS and DISCUSSION

  • The comparison should be limited to BAs that are detectable in all the samples. If you are also interested in other BAs you should develop a new analytical method that allows the proper quantification for all the analytes. In my opinion, considering all the analytes that were detectable in more than the 25% of the samples is not correct, because it is not representative of the population under study.
  • Line 207-209 should be moved to materials and methods section.
  • Please try to evaluate data considering the % variation of each BA at different time respect to the basal concentration. Try to understand if this %variation depends of the initial concentration of BAs in each subject under study. In my opinion it is not correct to consider the absolute variation of BAs at different times if the starting point (basal concentration) is so variable. If you consider the % variation is possible to highlight whether the increment or decrement of each bile acid is dependent on the initial state.
  • Please reports some information about the correlation between C0 C1 and C2 with the clinical parameters that you collected. Try to discuss how the subjects exhibiting metabolic syndrome characteristics are divided in these groups and how these observations are linked with the BAs composition and content.
  • Line 434-435 and Title “underscore metabolically healthy and unhealthy phenotypes”. Please give more evidences and discuss further this point. The text does not report strong evidence of what you are including in conclusion and title.

Reviewer 2 Report

Reviewer Comments

Summary of manuscript: This study administered a hypercaloric-high fat meal to healthy males. This was followed by measuring postprandial circulating levels of bile acids over 8 hours. They observed various patterns of bile acids in these participants. The postprandial patterns of bile acids were associated with “healthy” or “unhealthy” phenotypes.

Broad comments: This is a very detailed and thorough investigation regarding the production of various bile acids in the fasting and postprandial states. I provided my comments and suggestions below.

Minor comments:

Point 1: Abstract: Line 33: Consider changing hours to hour.

Point 2: Abstract: Line 37: Consider changing specie to species.

Point 3: Abstract: Lines 38-39: Conjugated and unconjugated were previously spelled out. Consider not using abbreviations here.

Point 4: Line 50: BA should be BAs, based on the abbreviation notation. Please check the abbreviations BA (singular) and BAs (plural) to be sure of the proper notations throughout the manuscript. For example, bile acid should be BA (line 92) and BAs should be BA (lines 95 and 99).

Point 5: Line 97: “Human” should be humans.

Point 6: Line 105: The sentence, “The potential effect of hypercaloric fat-rich diet…” should include “a” hypercaloric…

Point 7: Line 105: “western” was capitalized in the abstract. Perhaps be consistent.

Point 8: Line 123: “men” should be male, to be consistent with Table 1.

Point 9: Table 1, line 138: In the Table 1 footnote, please add abbreviation descriptions for ASAT, ALAT, TSH, and so forth.

Point 10: Line 142: FS refers to the fasting state, not fasting blood sample. Perhaps this can be indicated in the Figure 1 footnote.

Point 11: Figure 1, Line 149: This is a very nice figure. I would like to see Male volunteers, rather than “Men.” Additionally, I would prefer no smokers and no diabetics.

Point 12: Line 153: This is clinicaltrials.gov (no space between clinical and trials).

Point 13: Statistical analyses section, lines 186-202: As mention above, bile acid and bile acids were used in this section, whereas BA and BAs were indicated in the previous section. Perhaps these notations should be consistent.

Point 14: FS indicates the fasting state (line 190). However, the “fasting state” is used in this section (lines 192, 194, 197, and 198). Perhaps FS can be used in these instances.

Point 15: Lines 336-339: Please check this sentence. Are the terms “unlikely” and “nor” correct?

Point 16: Line 352: After “obese,” add subjects, individuals, or an associated term. You might want to remove “subjects” after “control” and place subjects after obese.

Point 17: Lines 435-437: Please review this sentence. I am not certain of the use of “prior appearance.” Should it state “prior to appearance”?

Author Response

Responses to R2

Comment: Broad comments: This is a very detailed and thorough investigation regarding the production of various bile acids in the fasting and postprandial states. I provided my comments and suggestions below.

Reponse: We thank this Reviewer for his/her positive comment on our study.

Minor comments:

Response: the manuscript has been corrected in strict accordance with Reviewer‘s suggestions

Reviewer 3 Report

Thank you for the opportunity to comment on this manuscript by Antonin Lamaziere and coauthors. I find the topic and research question both timely and interesting.

While bile acid metabolism has increasingly been associated with different components of metabolic syndrome, their diversity poses a challenge for their wider utilisation in the same way like interpreting lipoprotein concentrations does. The work by the Authors is thus a welcomed contribution.

I have the following comments:

Materials and Methods (page 3 of 24):

The original study protocol, as assessed from clinicaltrials.gov (NCT 03109067) and from ref#35 (Motte et al.2020) included more metabolic markers: insulin, HbA1C, HOMA-IR. They could have been used in the current work together with BMI and waist, for metabolic profiling. Furthermore, if platelet count was available, for instance FIB-4 index could have been estimated as a measure for liver fibrosis to be used as covariate. 

Study design (Figure 1. in page 5 of 24): The flow chart is clear and only makes me wonder: glucose and insulin/c-peptide were only measured at fasting state, I assume? Would have been great to be able to see their time course, regardless of the fact that their kinetics is faster and maybe had required 1-2 additional sampling points.

Study ethics (page 5 of 24):

The above mentioned study protocol, NCT 03109067, does not mention bile acids at all. While it is clear that extra sera from that protocol has been used for the current manuscript, referring to the original protocol is not precisely adequate, especially as obtaining written informed consent has been mentioned. 

Statistical analyses (page 6 of 24):

A comment of using Friedman test to assess changes over postprandial time course: this test is a purely parametric rank-order test that completely ignores the quantitative variation of the measured traits. While this is an advantage when the distributions are strongly skewed, it also disables a significant part of the time series analytics. 

Results:

It is maybe too strong to for instance call correlations btw GCA, GUDCA & C4 and TG as robust (page 11 of 24) without a possibility to see the actual scatter plots. I raise this b/c by looking later at the clusters C0-C2 (Figures 4 and 5, page 16 and 17 of 24), the N of samples in them varies widely. A way to help reader to compare the correlation coefficients could be to add N values to Table 3. Another way could be to add scatter plots to Supplements.

Another question from the Results: as, like the Authors mention, the interindividual variations of the BA concentrations are wide and their distributions non-Gaussian, is maximal change the best possible method to report their postprandial behaviour? It does not take into account differences in time course and is dominated by most extreme values. 

For multiple regression (page 11 of 24 and Table 4 in page 14 of 24), the actual regression model with dependent and independent variables should be given, and, if possible Rin addition to ß and p.

Discussion:

If one would like to summarise the result in a quick and dirty manner, one could say that a liquid high carbohydrate, high fat meal resulted in increases in primary BA’s driven by GCDCA, secondary and tertiary BA’s driven  GDCA and GUDCA, respectively as well as in sulfated forms of glycine and taurine BA’s, respectively. In total, conjugated BA’s increased and unconjugated BA’s decreased slightly. Interindividual variations was huge. Based on hierarchical clustering and PCA, three differing postprandial BA-clusters could be identified. Their reproducibility and physiological relevance remains to be seen.

In general, I understand well the difficulty of making connections between other traits and BA’s, but as this protocol provides information about age, anthropometrics, glucose and insulin etc, an attempt at least in discussion would be welcomed.

A minor detail: if all study subjects were either AA or GG homozygotes for CETP SNV rs708272, it would be interesting to see the Authors to comment in Discussion, whether this could have had any effect on the findings.

Round 2

Reviewer 1 Report

Authors have provided answers to all the questions and have improved the text where necessary.

I just only want to highlight that results regarding the association between BAs variation and metabolic syndrome markers are very preliminary (and this should be mention in the text, in my opinion) and future works on this subject should be based on a greater number of cases (samples) for each group in order to obtain more significative and useful results.